# Long Non-Coding RNAs in Sjögren’s Disease

**DOI:** 10.3390/ijms25105162

**Published:** 2024-05-09

**Authors:** Ondřej Pastva, Kerstin Klein

**Affiliations:** 1Department of Rheumatology and Immunology, Inselspital, Bern University Hospital, University of Bern, 3008 Bern, Switzerland; 2Department for BioMedical Research (DBMR), University of Bern, 3008 Bern, Switzerland

**Keywords:** long non-coding RNA, Sjögren’s disease, inflammation, autoimmunity

## Abstract

Sjögren’s disease (SjD) is a heterogeneous autoimmune disease characterized by severe dryness of mucosal surfaces, particularly the mouth and eyes; fatigue; and chronic pain. Chronic inflammation of the salivary and lacrimal glands, auto-antibody formation, and extra-glandular manifestations occur in subsets of patients with SjD. An aberrant expression of long, non-coding RNAs (lncRNAs) has been described in many autoimmune diseases, including SjD. Here, we review the current literature on lncRNAs in SjD and their role in regulating X chromosome inactivation, immune modulatory functions, and their potential as biomarkers.

## 1. Introduction

The dysregulation of non-coding RNA transcripts is a common characteristic of different human diseases, including autoimmune diseases [1], and typically occurs due to genomic, structural, and copy number variations; epigenetic modifications; or transcription factor alterations [2]. Given the broad regulatory functions of lncRNAs in transcription and translation, these changes have the potential to strongly impact normal cell activities, leading to cellular dysfunction.

Long, non-coding RNA (lncRNAs) are a large and highly diverse group of non-coding RNA molecules with a length of more than 200 nucleotides. They can be localized in the nucleus, the cytoplasm, or both, and can be divided into polyadenylated and non-polyadenylated lncRNAs [3]. A common classification of lncRNAs is based on their genomic localization relative to nearby protein-coding genes, which provides context and sometimes points to lncRNA function [4]. LncRNAs can be categorized into intergenic, intronic, promoter, enhancer, sense or antisense, and bidirectional lncRNAs. Some lncRNAs may also form circular structures [5]. The lack of a unique nomenclature for lncRNAs represents a hurdle in biomedical research, making the comparison of different studies difficult. Several attempts, led by the GENCODE consortium, the HUGO Gene Nomenclature Committee, and others, aim to provide a unique classification and nomenclature for lncRNAs [4,6,7]. Different databases, including NONCODE (http://www.noncode.org, accessed on 1 March 2024) [7], RNAcentral (https://rnacentral.org/, accessed on 1 March 2024) [8], and lncBook (https://ngdc.cncb.ac.cn/lncbook/, accessed on 1 March 2024; v2.0) [9] provide translation tools for lncRNAs.

LncRNAs play essential roles in various cellular processes, such as chromatin organization, gene transcription, the modification and splicing of nascent RNAs, the regulation of mRNA turnover, storage, and translation into protein [10]. The molecular mechanisms underlying the function of lncRNAs are highly heterogeneous, depending on their cellular location and interaction partners [10]. A single lncRNA might regulate multiple target genes as a molecular decoy that binds to transcription factors or micro-RNAs (miRNAs) to modulate their normal activity or availability. Through this mechanism, lncRNAs may influence signaling pathways and, subsequently, cellular responses to distinct environmental stimuli (Figure 1). The binding of lncRNAs to mRNA molecules can lead to intron retention, which promotes the alternative splicing of mRNAs [11]. Additionally, by their interaction with binding proteins, lncRNAs may serve as guides that promote protein-DNA interactions; proteins recruited by lncRNAs can inhibit or activate the transcription of targeted genes locally (*cis*) or remotely (*trans*). LncRNAs also operate as scaffolds to promote the formation of large multiprotein complexes, such as chromatin remodeling complexes, which can later be activated to affect gene expression [12]. Furthermore, a subset of lncRNAs contains open reading frames that allow their translation into functional peptides [13,14].

Although lncRNAs have been recognized as important regulators of transcription and translation, their function in health and disease is still largely obscure. Generally, lncRNAs are less conserved across species and expressed at lower levels than protein-coding mRNAs. This makes animal modeling and assessing their in vivo functions challenging. The most substantial advancements in lncRNA research are observed in the field of cardiovascular and neurodegenerative diseases [15,16]. To overcome the knowledge gaps regarding lncRNA function, the international FANTOM (Functional ANnoTation Of the Mammalian genome; https://fantom.gsc.riken.jp, accessed on 1 March 2024) consortium was launched, with FANTOM5 and FANTOM6 focusing on lncRNAs. While the FANTOM5 project aimed to provide a comprehensive lncRNA atlas of primary cell types derived from different species [17], the FANTOM6 project addressed the functional elucidation of lncRNAs [6]. Both projects have provided a rich data source for lncRNA research that is publicly available. Furthermore, FANTOM projects pointed once more to the large numbers of non-coding transcripts in the genome, with 70% being transcribed [18] but only 2% of transcripts being translated into proteins [19]. The current version of the NONCODE database (v6.0; accessed on 1 January 2024) has annotated more than 96,000 human lncRNA genes, generating more than 173,000 lncRNA transcripts [7]. Therefore, lncRNAs significantly outnumber coding genes (20,000, estimate), despite strict post-transcriptional surveillance, where only a subset of the non-coding transcripts possesses sufficient stability to persist in the cellular milieu [20].

Here, we review the current literature on lncRNA expression and function in Sjögren’s disease (SjD), an understudied autoimmune disease with huge unmet needs for patients and clinicians. SjD can occur either alone (formerly called “primary Sjögren’s syndrome”, pSjD) or in association with other autoimmune diseases (aSjD) [21]. It is characterized by chronic inflammation of salivary and lacrimal glands and the loss of secreting epithelial cells, resulting in severe oral and ocular dryness. Fatigue and chronic pain, together with severe dryness of mucosal surfaces, affect the majority of patients, with a tremendous impact on their quality of life. Clinically, patients present with very heterogeneous symptoms, ranging from mild to severe. In total, 30–50% of patients with SjD develop extra-glandular and systemic organ manifestations, including synovitis, cutaneous vasculitis, polyneuropathy, interstitial lung diseases, and nephritis, among others [22]. Classification criteria for SjD include objective measurements of diminished tear or saliva production, as well as the histological assessment of lymphocytic infiltrations into labial salivary glands (referred to as focus score) and/or the presence of anti-Ro/SSA autoantibodies [23]. SjD is still an underdiagnosed disease, in particular in its early stages and in patients presenting with atypical symptoms. Given that most patients show prominent glandular dysfunction at the time of diagnosis, biomarkers supporting early diagnosis are desperately needed. This would enable early therapeutic intervention and prevent the irreversible loss of epithelial cell secretory function [24]. 

Pathogenic mechanisms underlying SjD are still incompletely understood. Environmental factors, in particular, virus infections, might trigger the disease in genetically predisposed individuals [25,26,27]. Apoptosis of saliva-secreting epithelial cells often precedes salivary gland inflammation [28,29]. Their debris serves as a source of autoantigens for autoantibody formation [30]. Salivary gland lymphocytic infiltration with CD4+ T cell-rich and B cell-rich areas surrounding striated ducts and B cell activation are hallmarks of SjD [31,32]. Other immune cells, such as different dendritic cell subpopulations, might also be present [31]. The formation of ectopic germinal centers in tertiary lymphoid structures in some patients, representing sites of ectopic autoantibody production and the expansion of potential autoreactive B cell clones, is a risk factor for the development of mucosa-associated lymphoid tissue (MALT) lymphomas in patients with SjD [33]. 

Studying lncRNA function in autoimmune diseases has the potential to provide new insights into disease mechanisms and identify biomarkers for diagnosis, disease outcomes, or therapy response. LncRNAs are implicated in immune cell development, activation, and function [34]. Their dysregulation in autoimmune diseases might therefore impact disease development and outcomes and might reflect early changes in immune cell activation. In this review, we summarize the current literature on lncRNAs in SjD. We address their role in disease risk, with female sex and genetic variants being predisposing factors. Furthermore, we discuss lncRNAs as potential biomarkers for SjD and their immune regulatory roles and association with clinical characteristics assessed in patients with SjD. Most of the studies included in this review included patients with pSjD only. Since the current classification criteria for SjD do not distinguish between pSjD and aSjD [23], and this is also reflected in the recent literature, we address this aspect only in the context of Table 2. To make studies more comparable in the future, we translated lncRNA names into NONCODE gene IDs if available. 

## 2. Skewed X Chromosomal Inactivation

Female predominance is a common phenomenon in many autoimmune diseases, including SjD [35]. With a female-to-male ratio of 14:1, SjD has the highest prevalence in women compared to other autoimmune diseases (Table 1) [21]. Besides hormonal factors, the X chromosome itself might underly these discrepancies [35]. This theory is further confirmed by the increased prevalence of SjD in people with trisomy X (47, XXX) or Klinefelter’s syndrome (47, XXY) and the reduced risk of autoimmune diseases in women with Turner syndrome (45, X0) [36,37,38]. Among all chromosomes, the X chromosome encodes the highest number of immune system-related genes, including chemokine and interleukin receptors, co-stimulatory molecules, and Toll-like receptors [35,39]. To compensate for the X-linked gene dosage difference between females and males, the second X chromosome in females is silenced stochastically by a process called X chromosome inactivation (XCI) [40]. The master regulator of XCI is the long, non-coding RNA X inactive-specific transcript (XIST; NONHSAG054780.3), which is indispensable for the initiation of XCI [39,40]. Upon XCI, XIST expression starts from a genomic region on the X chromosome called the X inactivation center (XIC), from where XIST spreads across the chromosome, decorates it with XIST binding proteins, and triggers gene silencing [40,41]. XIST activation leads to profound changes in chromatin organization, characterized by a loss of activating histone marks and the formation of H3K27me3-positive heterochromatin, microscopically visible as Barr body [40,42].

The preferential silencing of either the maternal or paternal X chromosome is called skewed XCI, which has been described in different cell types in various autoimmune diseases (Table 1) [39]. Skewed X chromosome inactivation has been detected in the mesenchymal stromal cells of patients with SjD in comparison to cells from controls not fulfilling the classification criteria. In line with this finding, levels of XIST were decreased, paralleled by an increased expression of long, interspersed nuclear element-1 (LINE-1) retrotransposons [42]. The skewed XCI was accompanied by a redistribution of nuclear foci positive for the repressive histone mark H3K27me3, indicating a profound reorganization of the chromatin in cells derived from patients with SjD. The silencing of XIST in mesenchymal stromal cells recapitulated this phenotype. XIST expression was regulated by miR-6891-5p, a miRNA encoded in the human leukocyte antigen (*HLA*) locus. Given the strong genetic association of the *HLA* locus with the risk of SjD [25], this mechanism provides a potential link between genetic risk and female disease predominance. Skewed XCI in other relevant cell types and the impact on immune system-related X chromosome encoded genes in SjD remain to be investigated. 

During XCI, XIST interacts sequentially directly and indirectly with 81 binding proteins that form a ribonucleoprotein (RNP) complex that mediates silencing [48]. Several of these proteins serve as autoantigens and are exposed to the extracellular environment and the immune system once cells undergo cell death. Given the condensed form of XIST RNP complexes and the formation of a large polymer, the immunogenicity and potential for the activation of immune receptors is specifically increased in females [41]. Among the XIST binding proteins are small RNA binding exonuclease protection factor La (La/SSB) and heterogeneous nuclear ribonucleoprotein H1 (HNRNPH1) [41], two autoantigens identified in patients with SjD [49,50]. The hypothesis that an increased immunogenicity of XIST RNP complexes that is present only in females underlies female-biased autoimmunity has recently been evaluated in a pristane-inducible mouse model for systemic lupus erythematosus (SLE), in which XIST was overexpressed in male animals. Transgenic XIST expression in males reprogrammed T and B cells towards female-like changes and increased disease severity with a multi-organ phenotype [41]. The fact that serum reactivity towards antigens of XIST RNP complexes has been identified in patients with SLE and patients with systemic sclerosis (SSc) and dermatomyositis suggests that this is a generalizable mechanism in autoimmune diseases with female predominance.

## 3. Long, Non-Coding RNAs Link Genetic Risk to Cell-Type Specific Function

Genome-wide association studies (GWASs) have so far identified 22 genetic susceptibility loci (single-nucleotide polymorphisms; SNPs) that confer a risk of developing SjD and potentially affect more than 40 genes [25]. In 2015, Farh and coworkers revealed in their landmark paper that the majority of causative risk SNPs for 21 studied autoimmune diseases were located in cell-type and stimulus-specific enhancers [51]. Although this study did not include samples from patients with SjD and focused mainly on immune cells and omitted tissue-resident cell types, it underscored the importance of the non-coding genome as a carrier of the genetic risk for autoimmunity. Many risk SNPs for SjD overlap with those of other autoimmune diseases, and potentially credible SNPs are located in gene regulatory regions such as enhancers and promoters [25]. A subset of active enhancers, marked by activating H3K27 acetylation marks, can be transcribed in an RNA polymerase II-dependent process into so-called enhancer RNAs (eRNAs). These are a class of lncRNAs that facilitate the transcription of their target genes [52]. RNA sequencing (RNA-seq) of whole blood samples from patients with different autoimmune diseases detected more than 41,000 lncRNAs, including more than 2000 lncRNAs that had not been previously detected in patients with SjD. The novel lncRNAs were located near risk SNPs for different autoimmune diseases and overlapped with enhancers and super-enhancers of leukocytes, suggesting that these were eRNAs. An SjD-specific signature of enhancer activation and lncRNA expression in leukocytes was identified in approximately one-third of previously unknown and known lncRNAs that was not shared with other autoimmune diseases, including rheumatoid arthritis (RA), SLE, and others [53]. The study needs further validation in larger patient cohorts, given the low number of patients with SjD (*n* = 3) included, but potentially provides an important step towards a better understanding of SjD-specific activation of leukocytes. An additional overlay of the data set with promoter-capture Hi-C data would enable researchers to finally link the SNPs located in eRNAs and their target genes. In one such study, the overlay of SjD-risk SNPs with intergenic enhancer regulatory histone marks, together with promoter-capture Hi-C data and eQTL data, enabled the identification of potentially affected target genes in immune cells and salivary glands. Together, these analyses suggested potential genetic effects on immune cell function (CD247, NAB1, MIR146A, PRDM1, TNFAIP3, TYK2), inflammatory signaling (TNFAIP3, CRHR1, TYK2), cell survival and proliferation (CD247, MIR146A, PRDM1, TNFAIP3, TYK2), and cell stress (ATG5, CHMP6) [25]. 

## 4. Long, Non-Coding RNAs as Biomarkers

Biomarkers that support early SjD diagnosis, predict clinical outcomes, and direct therapy decisions in different patient subgroups are a matter of extensive research efforts. Blood-based biomarkers are of particular interest, given the easy and non-invasive method of sample collection. Another potential sample source for biomarker studies is saliva. Recently, lncRNAs were shown to account for more than 70% of differentially expressed RNAs in saliva-derived extracellular vesicles from patients with SjD compared to samples from patients with non-SjD sicca symptoms (nSS) [54].

Different approaches with the integration of multi-OMIC data sets, biological and clinical variables, symptoms, and machine learning have proven to be successful in sub-clustering SjD patient populations into three to four subgroups [55,56,57]. A re-evaluation of clinical trials based on a symptom-based sub-clustering of patients with SjD suggested drug efficacy in specific patient subsets [55,58], suggesting that future clinical trials could benefit from patient stratification tools. Given the tissue- and cell type-specific expression of lncRNAs, their inclusion into stratification tools and biomarker studies could provide a further asset. However, it is unlikely that the measurement of a single transcript will serve as a sole biomarker.

Ideal biomarkers would fulfill the following criteria: firstly, they should be reliably detectable in samples that can ideally be obtained easily and with minimal invasiveness; secondly, they should clearly distinguish assessed groups of patients; and, thirdly, identified potential biomarkers should be verified in independent sample cohorts. Given the large overlap of lncRNA expression levels between assessed groups in many studies, it is likely that the observed expression differences in patients with SjD and controls rather reflect differences in subgroups of patients. 

Most studies suggesting lncRNAs as biomarkers in SjD have compared their expression levels to those of healthy individuals and omitted a comparison with samples from other immune-mediated diseases with partially overlapping symptoms (Table 2). In addition, many studies were based on rather small sample sizes and need further validation. Another aspect that needs to be kept in mind is that biomarker studies based on whole blood or tissue samples might primarily reflect differences in cell composition rather than differential gene expression. 

A panel of five lncRNAs (linc0597, lnc0640, lnc5150, GAS5, lnc7074) measured in plasma samples exhibited 95% sensitivity and 85% specificity for the diagnosis of SLE (AUC = 0.966) compared to healthy controls. The same panel was used to distinguish patients with SLE from patients with SjD (AUC = 0.910, sensitivity: 67.5%, specificity: 96,8%), but it performed poorly in distinguishing patients with SLE from those with RA (AUC = 0.683, sensitivity: 67.5%, specificity: 66.7%). Plasma levels of linc0597, lnc5150, and lnc0640 were increased and levels of GAS5 and lnc7074 were decreased in patients with SLE compared to healthy controls or patients with SjD [59]. The same study identified higher plasma levels of lnc-DC in patients with SjD compared to patients with SLE [59]. These results were later validated in another study in which plasma levels of lnc-DC in patients with SjD and, in particular, those with immune thrombocytopenia were increased compared to those of healthy controls and patients with SLE or RA. The combination of lnc-DC plasma levels with anti-SSA/Ro and anti-SSB auto-antibody levels significantly enhanced diagnostic performance in discriminating patients with SjD from those with SLE and RA (AUC = 0.84, sensitivity: 78.5%, specificity: 89.91%) [60]. 

RNA-seq of whole blood samples derived from patients with SjD compared to samples from healthy individuals detected 1812 differentially expressed lncRNAs, with the majority being downregulated in SjD. A subanalysis of samples derived from patients that were either positive (SjD Ro^+^) or negative (SjD Ro^−^) for anti-Ro/SSA auto-antibodies identified 97 lncRNAs that were specific to SjD Ro^+^ when compared to healthy controls and 1114 lncRNAs that were unique in the SjD Ro^+^ subgroup [61]. At least some of the differences observed might be based on altered immune cell composition, which was addressed by the authors using a deconvolution analysis. Most cell types were similarly distributed among the two SjD subgroups and healthy controls, with the exception of M2 macrophages, which were increased in both SjD subgroups; monocytes were increased and CD4+ T cells were decreased in the SjD Ro^+^ group [61]. Linc01871, a lncRNA expressed in mature CD4+ and CD8+ T cells, class-switched memory B cells, plasma cells, natural killer cells, and hematopoietic progenitor cells was upregulated in SjD Ro^+^ and SjD Ro^−^ subgroups, with the highest expression found in the SjD Ro^−^ patients. Linc1871 was induced by interferon (IFN)-γ in Kasumi 3 (early myeloid) cells and regulated by calcineurin and TCR signaling in primary human T cells [61]. The expression of OAS123-AS1, MX1-AS1, GBP5-AS1, and NRIR was increased in SjD Ro^+^ patients but not in the SjD Ro^−^ subgroup [61], in line with the established higher IFN signatures in SjD Ro^+^ patients [62]. Increased levels of NRIR have also been described in peripheral blood mononuclear cells (PBMCs) of patients with SjD compared to healthy individuals [63]. Furthermore, NRIR has been identified as an IFN-inducible lncRNA, which was increased in CD14^+^ monocytes of patients with SSc compared to healthy controls [64]. The expression of OAS123-AS1, MX1-AS1, and GBP5-AS1 was also experimentally induced in Kasumi 3 cells by IFN stimulation, in which the lncRNA expression preceded the expression of the respective coding transcript [61]. This suggests a potential role of these lncRNAs in the regulation of coding IFN response genes. Since IFN signatures have been detected across different autoimmune diseases, including SjD, SLE, SSc, and RA [65], it is likely that additional IFN-inducible lncRNAs will be detected across different diseases. 

## 5. Immunomodulatory Properties of Long, Non-Coding RNAs 

Even though lncRNAs are known regulators of immune cell differentiation and function [66], functional evidence for their role in the pathogenesis of SjD is still limited. Only two studies have analyzed lncRNAs in salivary gland tissues so far (Table 2) [67,68]. One study detected 1243 differentially expressed lncRNAs in labial salivary glands from patients with SjD compared to patients undergoing labial salivary gland mucocele excision. Among them, eight upregulated lncRNAs were confirmed to be elevated in patients with SjD in an independent cohort and shown to positively correlate with clinical variables, such as disease duration, erythrocyte sedimentation rate (ESR), rheumatoid factors, and β2-microglobulin [67]. Cell types expressing these lncRNAs and their functional evaluation remain elusive and need further investigation. Such an analysis would be important to understand how altered lncRNA gene expression translates into decreased glandular function or increased lymphocyte activation. 

In a second study, lncRNA expression was evaluated in parotid tissues of patients with SjD or nSS and healthy individuals, enabling the authors to identify five lncRNAs that were common in SjD and nSS compared to healthy individuals (LINC00478, LOC101929072, LOC101929709, MIR205HG, RAD51-AS1), as well as lncRNAs that were specific to SjD (Table 2). LncRNAs shared between SjD and nSS correlated with coding genes enriched in immune-related biological processes, and M1 macrophage-relevant genes particularly correlated with LOC101929709. In total, 14 lncRNAs were identified in parotid tissues that discriminated patients with SjD from nSS and healthy individuals, with 10 being confirmed as increased in the labial gland tissues of patients with SjD (Table 2). Co-expression and correlation analysis suggested that these lncRNAs are potential regulators of immune cell infiltration, with CTA-250D10.23 being associated with chemokine signaling pathways. This was further underscored by the increased expression and high correlation of CTA-250D10.23 with C-C motif chemokine ligand 5 (CCL5) and C-X-C motif chemokine ligand 13 (CXCL13) in an independent validation cohort [68]. Monitoring serum levels of CXCL13 has been recently suggested as a biomarker for assessing the extent of salivary gland inflammation and lymphoid organization in patients with SjD [69]. Together, these data suggest that the inhibition of CTA-250D10.23 would potentially reduce salivary gland inflammation in SjD.

The differential expression of lncRNAs in PBMC from patients with SjD compared to healthy individuals was analyzed in several studies (Table 2) [63,70,71,72]. With this approach, several lncRNAs that were up- or downregulated in SjD were identified. However, conclusions regarding their role in the pathogenesis of SjD are difficult to draw since the cellular source of these lncRNAs has not been investigated, nor has a functional evaluation of at least selected lncRNAs been conducted. 

In one study, 199 differentially expressed lncRNAs were identified in PBMCs from anti-SSA/Ro-positive patients with SjD compared to healthy controls. Three lncRNAs (LINC00657, LINC00511, CTD-2020K17.1) were further validated and possessed miRNA targets that had previously been associated with different types of lymphoma [70]. A further in-depth evaluation of these lncRNAs, particularly including tissue samples of patients with SjD that develop lymphoma, would be needed to assess their potential role in lymphoma development or as predictive biomarkers. In addition, whether the inhibition of these lncRNAs could prevent the development of lymphoma should be tested. 

In another study, 1180 differentially expressed lncRNAs were detected in PBMC of patients with SjD compared to healthy controls, with GABPB1-AS1 and PSMA3-AS1 being the most upregulated and LINC00847 and SNHG being the most downregulated lncRNAs. Levels of GABPB1-AS1 correlated with the percentage of B cells and levels of IgG [71]. In a third study, 44 differentially expressed lncRNAs were identified in PBMCs from patients with SjD; TPTEP1-202 was confirmed to be downregulated and LINC00426, CYTOR, NRIR, and BISPR were upregulated compared to healthy individuals. While the expression of TPTEP1-202 negatively correlated with EULAR Sjögren’s syndrome disease activity index (ESSDAI) scores, CYTOR, NRIR, and BISPR showed a positive correlation [72]. 

Additional studies on specific peripheral blood-derived cell types provided more meaningful insights into the role of lncRNAs in SjD. PVT1, an lncRNA that has previously been identified to be upregulated in the labial salivary gland tissues of patients with SjD [67], was specifically upregulated in the CD4+ T cells of patients with SjD compared to healthy individuals [73]. PVT1 controlled the expression of the transcription factor c-Myc, a master regulator of T cell expansion and differentiation, upon antigen stimulation, and programmed CD4+ T cells towards glycolysis [73]. Neat1, another lncRNA that is increased in the CD4+ T cells of patients with SjD, positively regulated the phosphorylation of the p38 and ERK1/2 pathways and, thus, the production of C-X-C motif chemokine ligand 8 (CXCL8) and the tumor necrosis factor (TNF) [74]. Increased levels of Neat1 have additionally been identified in monocytes from patients with SLE [75].

TMEVPG1 (NEST, IFNG-AS1) was shown to be upregulated in CD4+ T cells from patients with SjD, with a higher expression in patients positive for anti-Ro/SSA or ANA auto-antibodies. Levels of TMEVPG1 correlated with the proportion of T helper 1 (Th1) cells and the expression of IFN-γ and the transcription factor T-bet. The silencing of TMEVPG1 in CD4+ T cells reduced numbers of Th1 cells and IFN-γ expression [76]. Similar findings were described in patients with Hashimoto’s thyroiditis [63]. Moreover, transcript levels of TMEVG1 were also elevated in the PBMCs of RA patients [77], indicating a potential broader relevance of TMEVG1 dysregulation in autoimmune disorders.

The dysregulation of B cells and upregulation of the interferon (IFN) signaling pathway are characteristics of SjD [32]. The lncRNA LINC00487 was found to be upregulated in four peripheral blood B cell subsets of patients with SjD compared to healthy individuals. Expression levels of LINC00487 positively correlated with the expression of IFN-signature genes and disease activity. In line with these findings, LINC00487 expression was induced by IFN-α [78]. LINC00487 was additionally identified as a differentially expressed lncRNA in two independent studies analyzing parotid tissue and whole blood samples of patients with SjD [61,68]. Furthermore, LINC00487 has been described in B cell development, where it was co-expressed with germinal center B cell markers [79], providing a potential link between this lncRNA and B cell autoreactivity in SjD. Another IFN-α-inducible lncRNA is MALAT1 (NONHSAG008675.3), which was similarly induced in PBMCs from patients with SjD and controls [80]. MALAT1 has been shown to be a negative regulator of IFN-α production in the innate response to viral infection. Its expression was aberrantly decreased in PBMCs of patients with SLE and inversely associated with the expression of IFNA [81]. Whether this association is also present in patients with SjD remains to be elucidated. Further studies would be needed to evaluate whether inhibition of the above-mentioned lncRNAs would decrease glandular inflammation by decreasing CD4+ T cell expansion and B cell activation, the hallmarks of SjD [31,32]. 

## 6. Conclusions

Several studies have identified differentially expressed lncRNAs in patients with SjD compared to healthy individuals or disease controls. Some of these lncRNAs have been identified in more than one study, suggesting consistent expression changes in SjD. Since the upregulation of first lncRNAs in salivary gland tissues and peripheral blood has been described, it would be interesting to further investigate whether these lncRNAs could be measured in peripheral blood as a surrogate for lymphocyte infiltration into salivary gland tissues. his premises cell-type specificity of such lncRNAs to enable a conclusion on pathogenic events within the tissue and would enable to longitudinally monitor the inflammatory state within the salivary glands. The sole descriptive nature of many studies points to the need for future functional studies of identified lncRNAs. The available biomarker studies need further validation, particularly with the inclusion of samples from other autoimmune diseases. Together, such studies will pave the way towards a better understanding of the role of lncRNAs in the pathogenesis of SjD. 

**Table 2 ijms-25-05162-t002:** LncRNAs in Sjögren’s disease.

lcnRNA	NONCODE Gene ID	Origin	Method	No. of Patients (Discovery Cohort)	Validation Method	No. of Patients (Validation Cohort)	Up- or Downregulation (SjD versus Control)	Reference
lnc-DC	n.a.	blood plasma	qPCR	149 pSjD, 50 SLE, 50 RA, 109 HC	n.v.		up	[60]
LINC01871	NONHSAG026917.2	whole blood	RNA-seq	50 SjD ^§^ (27 Ro^+^, 23 Ro^−^), 27 HC	qPCR	22 SjD (14 Ro^+^, 8 Ro^−^), 24 HC	up in SjD (Ro^+^ and Ro^−^)	[61]
NRIR	NONHSAT068918.2						up in SjD Ro^+^	
OAS123-AS1	n.a.							
MX1-AS1	n.a.							
GBP5-AS1	n.a.							
TPTEP1-202	n.a.	PBMCs	RNA-seq	5 pSjD, 5 HC	qPCR	16 pSjD, 6 HC	down	[72]
LINC00426	NONHSAG013150.2						up	
CYTOR	NONHSAG077527.2							
NRIR	NONHSAG026902.2							
BISPR	NONHSAG025088.3							
LINC00657	NONHSAG031696.3	PBMCs	microarray analysis	8 pSjD, 8 HC	qPCR	n.a.	down	[70]
LINC00511	NONHSAG022655.3						up up	
CTD-2020K17.1	n.a.							
GABPB1-AS1	NONHSAG016861.3	PBMCs	RNA-seq	4 pSjD, 4 HC	qPCR	30 pSjD, 15 HC	up	[71]
PSMA3-AS1	NONHSAG015097.3							
TMEVPG1 (NeST, IFN-γ-AS1)	NONHSAG011599.2	CD4^+^ T cells	qPCR	20 pSjD, 10 HC	n.v.		up	[76]
NEAT1	NONHSAG008670.3	CD4^+^ T cells	qPCR	20 pSjD, 10 HC	n.v.		up	[74]
PVT1	n.a.	CD4^+^ T cells	qPCR	25 SjD, 25 HC	n.v.		up	[73]
LINC00487	NONHSAG026900.2	B-cells	microarray analysis	6 pSjD, 6 HC	qPCR	14 pSjD, 12 HC	up	[78]
CTA-250D10.23	n.a.	parotid gland	microarray analysis	19 pSjD, 20 HC, 20 nSS	microarray analysis (labial SG) *		up	[68]
KIAA0125	n.a.							
LOC100505812	n.a.							
BZRAP1-AS1	n.a.							
LINC01215	NONHSAT194282.1							
PSMB8-AS1	NONHSAT108940.2							
ITGB2-AS1	NONHSAT082896.2							
LOC100505549	n.a.							
LOC101929272	n.a.							
ENST00000420219.1	n.a.	labial SG	microarray analysis	4 pSjD, 4 C (mucocele excision)	qPCR	30 pSjD, 16 C (mucocele excision)	up	[67]
ENST00000455309.1	NONHSAG028948.3							
NR_002712	n.a.							
ENST00000546086.1	NONHSAG011610.3							
n340599	NONHSAG041352.2							
TCONS_l2_00014794	n.a.							
n336161	n.a.							
lnc-UTS2D-1:1	n.a.							

Multiple long, non-coding RNAs (lncRNAs) described with roles in the pathogenesis of Sjögren’s disease. C, control; HC, healthy control; IFN, interferon; n.a., not available; n.v., not validated; qPCR, quantitative polymerase chain reaction; PBMC, peripheral blood mononuclear cells; pSjD, primary Sjögren’s disease; RA, rheumatoid arthritis; Ro^+^, positive for anti-Ro/SSA-; Ro^−^, negative for anti-Ro/SSA-; SG, salivary gland; SLE, systemic lupus erythematous; * patients were diagnosed based on [82] and scored into severity groups based on [83]; ^§^ including patients with pSjD and associated SjD.

## Figures and Tables

**Figure 1 ijms-25-05162-f001:**
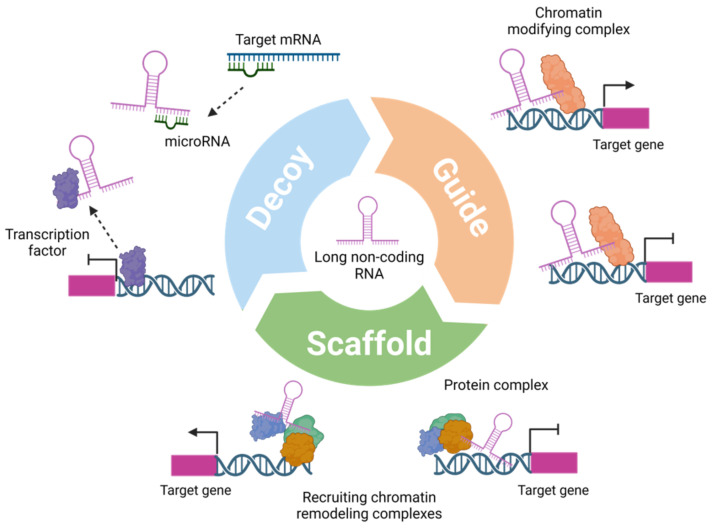
Roles and mechanisms of lncRNAs. LncRNAs can function as decoys, binding transcription factors, other proteins, or miRNAs, which they sequester from chromatin (upper left). LncRNAs can guide transcription factors to specific genomic sites, orchestrating gene expression regulation (upper right). Additionally, they act as scaffolds, facilitating the formation of chromatin remodeling complexes (lower part).

**Table 1 ijms-25-05162-t001:** Prevalence of different autoimmune diseases in female and male individuals.

	SjD	SLE	RA	SSc
**Prevalence** **female/male**	14:1 [21]	9:1 [43]	3:1 [44]	3–8:1 [45]
**Skewed XCI**	present [42]	present [46]	present [47]	present [45]

SjD: Sjögren’s disease; SLE: systemic lupus erythematosus; RA: rheumatoid arthritis; SSc: systemic sclerosis; XCI: X chromosome inactivation.

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
