# Peer review of "Long Non-Coding RNAs in Sjögren’s Disease"

_ijms, 2024, doi:10.3390/ijms25105162_

Round 1

Reviewer 1 Report

Comments and Suggestions for Authors

I think it is a big problem the authors made discussion on comparing SS with other rheumatic diseases such as SLE and RA, without discriminating primary Sjögren’s Syndrome (SS) from secondary SS. As the authors described in line 85 to 86, is classified into primary SS and secondary SS based on the presence or absence of other rheumatic diseases including RA and SLE i.e. primary SS is a specturum clinically different from secondary SS (Please note that the authors of Ref.55 stated they investigated PRIMARY SS). I think the authors keep primary SS is different from secondary SS in mind and should re-write review whole discussion.

Author Response

Please find our response in the uploaded pdf file.

Reviewer 2 Report

Comments and Suggestions for Authors

In the current review, the authors summarize current knowledge about the involvement of long noncoding RNAs in Sjogren’s disease, an autoimmune disease. The authors discussed lncRNAs with roles in disease development and lncRNAs serve as potential biomarkers for the disease. I have few suggestions here, which I hope can help improve the quality of the review.

1.       In the introduction section, the portion for introducing the molecular mechanisms of lncRNAs in gene regulation is well organized and detailed. I would suggest adding some paragraphs focusing on the current roles of lncRNAs in autoimmune diseases, current limitations in SjD diagnosis and treatment, and why lncRNAs may serve as good therapeutic target/biomarker for SjD.

2.       Overall, the structure of section 2 through section 5 is a bit confusing as the category of lncRNAs under each section are not very clear. I would suggest the authors reorganized the contents in these sections and develop a more clearly classification of these lncRNAs in SjD. For examples, it might be helpful to separate the lncRNAs based on their roles in disease risk assessment, development, progression, or treatment outcomes.

3.       Only one table was given in the review, which correspond the content in section 5. I would suggest the authors generate one table for each section.

4.       In general, the discussion part is missing for section 2 through section 5, including the conclusion section. I would suggest the authors adding more conclusive sentences or paragraphs to better summarized the content in each section.

Comments on the Quality of English Language

N/A

Author Response

Please find our response in the uploaded file.

Reviewer 3 Report

Comments and Suggestions for Authors

The manuscript "Long non-coding RNAs in Sjögren's Disease" by OndÅ™ej Pastva et al. is a complete literature review on a largely unexplored subject. The authors address the role of lncRNAs in X chromosome inactivation, immune modulation, and their potential as biomarkers of Sjögren's Disease, highlighting the descriptive nature of many studies and the need for more future studies that shed insight into the functional role of lncRNAs in its pathogenesis. 

In Table 2, the authors describe all the studies of lncRNAs associated with Sjögren's Disease and include their origin, which varies from labial SG, T and B cells, plasma, PBMCs, parotid gland, and whole blood. Could the authors further address the importance of these lncRNAs' origin for Sjögren's Disease pathogenesis and how the sample type impacts their potential roles as biomarkers as part of their conclusions?

Comments on the Quality of English Language

Line 19: remove "in"

Line 20: add a comma between "genomic" and "structural"

Line 45: response -> responses

Line 64: there -> their

Line 120: Add a comma between "co-stimulatory molecules" and "toll-like receptors"

Line 123: Use the lncRNA abbreviation instead of the full name 

Line 147: erythematousus -> erythematosus

Line 212: add "the" between "fulfill" and "following criteria."

Line 285: as -> are

Line 324: the word "expression" is repeated

-Check all apostrophes, as they are all backward.

-Define abbreviations when first mentioned (IFN, PBMC, ESR, nSS, CCL5, CXCL13, ESSDAI, CXCL8, TNF). Also, SLE: Systemic lupus erythematosus, RA: rheumatoid arthritis, and SSc: systemic sclerosis are defined in Table 1 but not within the manuscript's text. 

-Revise the spelling of up-regulated to be consistent throughout the manuscript (up regulatedupregulatedup-regulated?)

Reviewer 4 Report

Comments and Suggestions for Authors

The manuscript entitled “Long non-coding RNAs in Sjögren's Disease” is a narrative review on the role of lncRNAs in molecular pathogenesis of Sjögren's Disease and on their biomarker potential. The topic of the manuscript is within the scope of the journal and would be interesting to the readership focusing on autoimmune disorders and gene regulation mechanisms that involve lncRNAs. It offers an up-to-date overview on the main findings and authors made efforts to highlight certain results that require additional confirmation and validation, as well as to point out areas that are still largely unknown.

The manuscript is very well written, well structured, the overall presentation is detailed, clear and informative, while the conclusions are well-supported by the reviewed findings. There are merely some minor issues that should be resolved and some corrections are needed:

- line 11: “patients SjD” should be “patients with SjD”;

- line 19: delete extra “in”;

- lines 54-55: This reference [9] is relatively old one. There are known functional peptides translated from lncRNAs, like in 10.3389/fonc.2021.777374 or 10.3389/fonc.2020.622294, so the ability of lncRNAs to encode functional peptides is not in question;

- The sentence in lines 64-66 is also not entirely correct. I suggest that the authors add “generally” at the beginning of this sentence, since there are some lncRNA which are very abundant, like MALAT1, and are expressed at higher level compared to some mRNAs;

- The connection between lncRNA and SjD is not well demonstrated in the corresponding paragraphs of the Introduction section. I suggest that the authors include one paragraph at the end of Introduction that will explain the main reasons for conducting the review with this particular topic and what is the exact novelty compared to previous similar reviews, like 10.3390/genes12060903;

- line 222: another important criteria for ideal biomarker is that the acquisition of samples is minimally invasive;

-line 250: delete “in”;

- lines 284-286: this sentence needs to be rephrased. Either replace “as” with “are”, or add “act” after “lncRNAs”.

- line 313: delete spacing between “up” and “regulated”.

Comments on the Quality of English Language

Minor editing needed.

Round 2

Reviewer 2 Report

Comments and Suggestions for Authors

My questions and suggestions have been addressed in the revised manuscript.